# Effect of postural threat on motor control in people with and without low back pain

**Meta H. Wildenbeest** [1,2] *, **Henri Kiers** [1,2], **Matthijs Tuijt** [1], **Jaap H. van Dieën** [2]

**1** Institute for Human Movement Studies, HU University of Applied Sciences, Utrecht, The Netherlands,
**2** Department of Human Movement Sciences, Amsterdam Movement Sciences, Vrije Universiteit Amsterdam, Amsterdam, The Netherlands

☯ These authors contributed equally to this work.
\* meta.wildenbeest@hu.nl

**Data Availability Statement:** Data available at Open Science Framework: Dataset Lumbar Movement Patterns repetitive seated reaching. https://doi.org/10.17605/OSF.IO/9XSNP

## Abstract

### Introduction

Negative pain-related cognitions are associated with persistence of low-back pain (LBP), but the mechanism underlying this association is not well understood. We propose that negative pain-related cognitions determine how threatening a motor task will be perceived, which in turn will affect how lumbar movements are performed, possibly with negative long-term effects on pain.

### Objective

To assess the effect of postural threat on lumbar movement patterns in people with and without LBP, and to investigate whether this effect is associated with task-specific pain-related cognitions.

### Methods

30 back-healthy participants and 30 participants with LBP performed consecutive two trials of a seated repetitive reaching movement (45 times). During the first trial participants were threatened with mechanical perturbations, during the second trial participants were informed that the trial would be unperturbed. Movement patterns were characterized by temporal variability (CyclSD), local dynamic stability (LDE) and spatial variability (meanSD) of the relative lumbar Euler angles. Pain-related cognition was assessed with the task-specific 'Expected Back Strain'-scale (EBS). A three-way mixed Manova was used to assess the effect of Threat, Group (LBP vs control) and EBS (above vs below median) on lumbar movement patterns.

### Results

We found a main effect of threat on lumbar movement patterns. In the threat-condition, participants showed increased variability (MeanSD$_{flexion-extension}$, p<0.000, η² = 0.26; CyclSD, p = 0.003, η² = 0.14) and decreased stability (LDE, p = 0.004, η² = 0.14), indicating large effects of postural threat.

**Funding:** This work was supported by the Dutch Organization for Scientific Research: Nederlandse Organisatie voor Wetenschappelijk Onderzoek (NWO), The Hague, The Netherlands [grant number 0.23 0.12 0.25]. The funders had no role in study design, data collection and analysis, decision to publish, or preparation of the manuscript.

**Competing interests:** The authors have declared that no competing interests exist.

## Conclusion

Postural threat increased variability and decreased stability of lumbar movements, regardless of group or EBS. These results suggest that perceived postural threat may underlie changes in motor behavior in patients with LBP. Since LBP is likely to impose such a threat, this could be a driver of changes in motor behavior in patients with LBP, as also supported by the higher spatial variability in the group with LBP and higher EBS in the reference condition.

## Introduction

Low back pain is the number one cause of disability worldwide and consequently causes high healthcare and socio-economic costs [1, 2]. For most patients with LBP, it is currently not possible to identify a specific nociceptive source. The common recurrence of LBP-episodes is only moderately affected by treatment, such as motor control exercise (MCE) [3, 4]. The development of more effective treatment methods is challenged by a lack of knowledge regarding the mechanisms underlying chronicity of LBP [5].

Established risk factors for the development of chronic LBP (CLBP) include psychological factors, such as fear of movement and catastrophizing [6, 7]. It has been suggested that these negative pain-related cognitions may alter motor control, by means of the patient's concern of losing control over trunk movements [8]. In addition, motor control changes are thought to have long-term adverse effects, thereby potentially adding to the chronicity of LBP [8, 9]. If these assumptions are correct, the association between pain-related cognitions and pain persistence may be mediated by altered motor control. As a first step to investigate these possible associations, we focused in this study on the immediate effect of a threat to perturb trunk movement and control and its interaction with pain-related cognitions on motor control.

The association between pain-related cognitions and motor control, quantified as local dynamic stability, has been studied previously with contradictory results. Motor control is defined as the way in which the nervous system controls posture and movement to perform a specific task, including all motor, sensory and integrative processes [10]. In this and our previous study [24] we used variability and local dynamic stability (LDS) as, respectively, linear and non-linear outcome measures to quantify the quality of this motor control. As described by Dingwell and Marin [11], spatial variability is the mean standard deviation (MeanSD) of lumbar angles of a repeated movement. Temporal variability (CyclSD) is quantified as the standard deviation of cycle times (CycleSD) [12]. As also described in our previous studies [9, 13], LDS is the inverse of how fast movement patterns are corrected, after very small perturbations, for example caused by eye blinking or breathing. LDS is commonly expressed by the local divergence exponent (LDE). The LDE is the mean logarithmic rate of divergence between neighboring trajectories in lumbar kinematics state space [11, 12, 14–16]. Individuals with experimentally induced pain displayed both, increased [17] and decreased local dynamic stability [18] of lumbar movement patterns in relation to pain and pain-catastrophizing. A contributing factor to these conflicting results could be that pain-related cognitions were measured with questionnaires addressing pain-related cognitions as a trait [17, 18] e.g. 'the Pain Catastrophizing Scale' (PCS) [19] or 'the Pain Anxiety Symptoms Scale' (PASS) [20] However, pain-related cognitions are dynamic mutable, influenced by the task at hand and its current context. Consequently, pain-related cognitions may show more clear relations with

motor control if a task-specific measurement tool is used [9, 21, 22]. In line with this, we found greater lumbar movement variability in those participants with LBP, who expected task demands to impose a threat of pain or reinjury of the back, while no association with outcomes of commonly used questionnaires on pain-related cognitions were found. Additionally, correlations between task-specific expected threat of pain or reinjury of the back and scores on clinical questionnaires on pain related cognitions were moderate at best [9]. We suggest that the interaction of LBP and expected threat of pain or reinjury of the back can be conceptualized as a perceived threat, which is context specific. LBP will affect the valuation of imposed threat of pain or reinjury of the back, due to an increase in perceived risk of injury [23], and hence the interaction of these two factors determines the perceived threat, which in turn affects motor control.

In the present study, we experimentally manipulated perceived threat to assess a causal relationship with motor control changes. An externally imposed threat would be expected to add to or interact with LBP-status and expected threat of pain or reinjury of the back. Therefore, we assessed the immediate effect of postural threat on lumbar movement patterns, during repetitive seated reaching, in people with and without LBP. We chose repeated seated reaching because this movement is a more demanding task for the back than walking, which was the experimental task in most previous studies. Additionally, seated reaching provides the opportunity to focus on the movements of the lumber spine largely eliminating effects of leg movements [24]. As experimentally induced LBP has been shown to decrease local dynamic stability in healthy young men [17] and we previously reported an increase of spatial variability in people with LBP who expected task demands to impose a high threat of pain or reinjury of the back [9], we hypothesized that in both, people with and without LBP, postural threat increases variability and decreases local dynamic stability of lumbar movement patterns.

## Materials and methods

### Participants

From September 2020 till April 2021, thirty participants with LBP and 30 back-healthy participants (matched by age and sex) participated in this study (Table 1). They had been recruited through word of mouth by the researchers and students involved. Additionally, physical- and exercise therapists were asked to invite patients, who met the inclusion-criteria. In- and exclusion criteria were assessed using a self-administered questionnaire supervised by a trained exercise therapist. Participants with LBP were included if: (1) they had experienced more than one episode of non-specific LBP or continuous non-specific LBP within the last two years; (2) the duration of an episode of LBP was at least two weeks; (3) the pain intensity was affected by posture or movement. The inclusion criterion for back-healthy participants was to be free from episodes of LBP for at least 2 years. Exclusion criteria for the LBP-group were positive red flags and for both groups (1) perceived balance problems; (2) BMI over 30 in combination with high abdominal circumference (males > 102 cm, females > 88cm); (3) any systemic (e.g., diabetes mellitus), cardiovascular or neurological pathology, infections, medication which might influence movement (antidepressants, analgesics, tranquillizers), earlier spine surgery, pregnancy, significant musculoskeletal injury in the past six months or respiratory ailments [13]. All participants provided written informed consent, prior to participation. The protocol had been approved by the ethical committee of the Faculty of Behavioral and Human Movement Sciences, VU University of Amsterdam (VCWE-2020-070).

**Table 1. Participant characteristics.**

|  | LBP (N = 30) Sex (M/F): 11/19 | Back-Healthy (N = 30) Sex (M/F): 11/19 | F ratio | p value |
|---|---|---|---|---|
|  | Mean (SD) | Mean (SD) |  |  |
| Age (year) | 32.1 (13.6) | 32.0 (13.5) |  |  |
| Height (cm) | 1.79 (0.10) | 1.79 (0.08) | 2.69 | 0.1 |
| Weight (kg) | 74.7 (14.1) | 74.7 (10.3) | 2.67 | 0.1 |
| ODI/50 | 15.7 (12.7) |  |  |  |
| SBST/9 | 1.6 (1.5) |  |  |  |
| LBP intensity at day of testing/10 | 2.4 (2.1) |  |  |  |
|  | Median (IQR) | Median (IQR) | U | p value |
| EBS$_{postural\ threat}$/10 | 3 (2.00–4.25) | 2 (2.00–4.00) | 363 | 0.19 |
| EBS$_{reference}$/10 | 2 (1.75–4.00) | 2 (1.00–3.00) | 376 | 0.26 |
| PCS/52 | 13 (8.75–18.25) | 13 (7.00–15.50) | 417 | 0.63 |
| PASS/200 | 45 (35.75–63.50) | 48 (38.75–72.00) | 384.5 | 0.33 |

ODI Oswestry Disability Index, SBST StarT Back Screening Tool, EBS Expected Back Strain, PCS Pain Catastrophic Scale,
PASS Pain Anxiety Symptoms Scale

## Clinical characteristics

Participants with LBP completed the StarT Back screening tool (SBST) [25], where a score 0–3 indicates low risk of psychosocial prognostic risk factors and a score ≥ 4 indicates high risk. Additionally, the Oswestry Disability Index (ODI) [26] was completed, with a minimum score indicating no disability and the maximum score indicating 100% disability. Patient-reported Pain intensity at day of testing was registered, by a verbal numerical rating scale (NRS) (0 = no pain, 10 = worst imaginable pain).

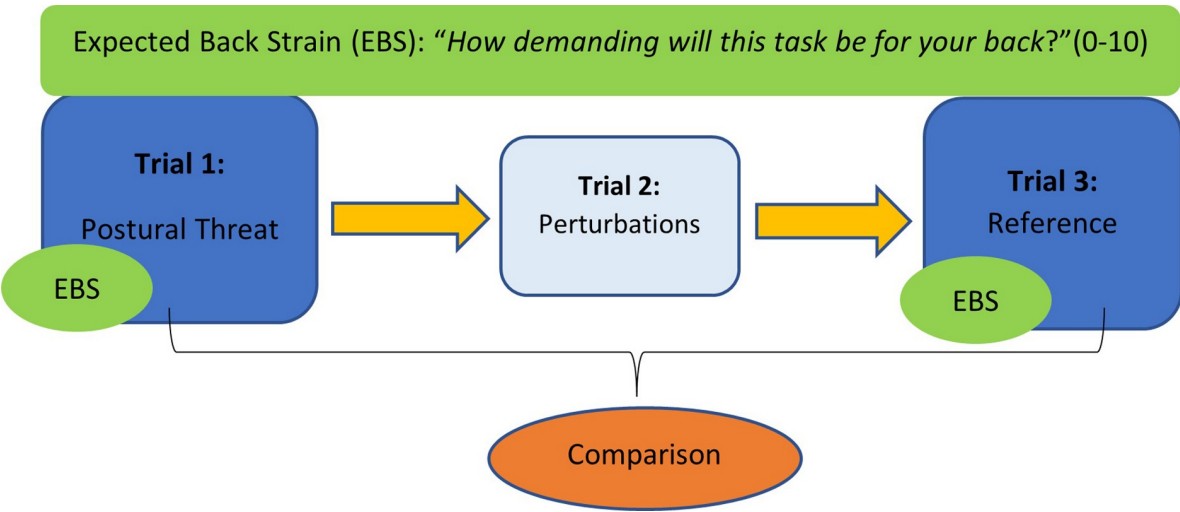

**Fig 1. Experimental design.** Each trial consisted of 45 repetitive seated reaching movements.

## Materials

As described previously [13], a custom-made chair, without back rest and arm supports, was rigidly attached to a DynSTABLE (Motek Medical Amsterdam, Netherlands). A motion capture system consisting of four Vicon Bonita3 cameras (VICON-612 system, Oxford Metrics, UK) was used to track reflective markers, and sampled with D-Flow software at approximately 100 samples/s (Motek Medical Amsterdam, Netherlands). To assess lumbar movement patterns, two clusters of three markers were used. These clusters were fixed to the spinous processes of T8 and S1 using adhesive tape [13].

## Experimental procedure

Participants performed three trials with a pause of a few minutes in between (Fig 1). To prevent fatigue, participants could step off the DynSTABLE between trials, if necessary, similar to the procedure in a study on test-retest reliability of the variables assessed in the present study [13]. Each trial consisted of 45 slightly asymmetric reaching movements, performed seated, at self-selected pace, with one arm crossed in front of the chest (Fig 2). For familiarization, participants practiced the reaching movement five times, before each trial. The task started from an upright posture and required participants to repetitively reach for the button of a joystick positioned at knee level in front of them, at a distance of 125% of their arm length. To control movement amplitude, the joystick button had to be pressed with the fingers of the dominant hand (Fig 2) [13]).

To impose a postural threat, participants were warned for mechanical perturbations prior to the first trial. Participants were informed that the chair, would move unexpectedly in a random direction, that this mechanical perturbation could be intense, and that they were allowed to grab the handrails when losing their balance. They were instructed to recover as quickly as possible and keep on reaching to finish the trial. However, during the first trial no perturbations were applied. In the second trial, mild mechanical perturbations were actually administered. The third trial was used as a reference; participants performed the task knowing that no perturbation would occur. To investigate the effect of postural threat on the movement patterns, we compared only the first and the third trial.

## Pain-related cognitions

To quantify the task-specific pain-related cognition, we established the expected back strain by asking "How demanding will this task be for your back?". Participants responded to this question by means of an 11-points color-marked scale (0–10), based on the RPE-Borg scale, further on referred to as the 'Expected Back Strain' (EBS) (S1 File) [27]. To assess the EBS associated with the first trial, participants completed this EBS prior to the first trial, but after being informed about the threat of mechanical perturbations during this trial. This EBS-scale was completed again before the reference trial. As trait characteristics of pain-related cognitions, we assessed pain catastrophizing (Pain Catastrophizing Scale (PCS)) [19] and pain-related fear (Pain Anxiety Symptoms Scale (PASS)) [20], prior to the instruction regarding trial 1.

## Lumbar movement patterns

**Joint kinematics.** To exclude transients in the data, the final 40 of the 45 repetitions of the first and third trial were selected for analysis. Data were cubic spline interpolated to 100Hz, to account for missing samples and correct for small fluctuations in sample rate caused by D-flow software recording at 102/103 Hz. Segment orientations were computed in the global axis system. Subsequently, relative orientations between thorax and pelvis were determined and

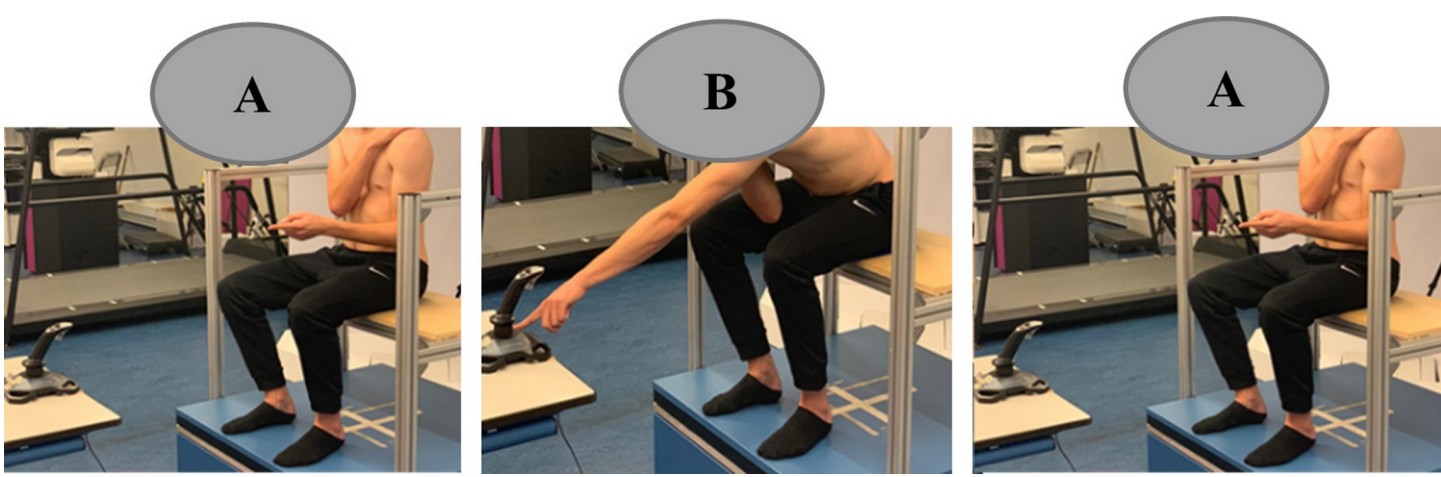

**Fig 2. Overview of the task performed by the participants.** Starting position–upright sitting (A), forward reaching to a distance of 1.25 x length of the upper limb to reach the target (B), and starting position again (A).

decomposed into lumbar angles using Euler decomposition in the order flexion/extension, lateral-bending, axial-rotation [13].

**Variability.** The time series of the lumbar angles were divided into cycles, based on peak detection of the most forward sagittal plane orientation of the thorax (T8) in each cycle. Temporal variability was quantified as the standard deviation of the cycle durations (CyclSD). For spatial variability, lumbar angle data for each cycle were normalized to 101 samples (0–100%) by spline interpolation for all three axes. Cross-correlation was used to optimally align all repetitions. Spatial variability was then calculated as the average of the standard deviations at all normalized time points across the cycles (MeanSD) [11].

**Local dynamic stability.** We operationalized local dynamic stability by means of the local divergence exponent (LDE). As the number of samples affects the LDE, lumbar angle time series were normalized to a fixed number of data points (300 times the number of cycles), using cubic spline interpolation [28]. The three lumbar angles were used to reconstruct a 6-dimensional state-space, with one 30-samples (10% of the average number of samples per cycle) time-delayed copy. To minimize effects of noise, we tracked divergence between kinematic states evolving from each data point and its 15 nearest neighbors [16]. Divergence curves were logarithmically transformed and averaged over the nearest neighbors per reference point and over reference points. LDE was determined as the slope of the best fitting line over the first 0.25 cycle of the resulting divergence curve [15]. The algorithm used is available at https://zenodo.org/record/4681213 [29].

**Amplitude and velocity.** Maximum and minimum lumbar angles per cycle were determined, for all three directions. Subsequently, mean maximum and mean minimum angle were subtracted to determine the range of motion. Velocity was calculated by dividing the number of repetitions by the total time of the trial.

## Statistical analyses

Statistical analyses were performed with IBM SPSS Statistics 25 software. Independent t-tests were used to test for differences between people with and without LBP, concerning height and weight. A Mann-Whitney U test was used to test for possible differences concerning pain-related cognitions between groups. To assess whether the task instruction affected EBS, EBS-scores before the threat and reference conditions were compared using a Wilcoxon signed-

rank test. In addition, the correlation between EBS-scores before both trials was calculated, to assess whether subjects could consistently be classified based on either of these measurements. High and low levels of EBS were based on a median split (low level of EBS≤3, high level of EBS≥4). The effect of threat on lumbar movement patterns was assessed with a three-way mixed MANOVA, using threat, dichotomized EBS and group as independent variables. Separate analyses were done for variability and stability on one hand, to test our primary hypothesis, and movement amplitude and velocity on the other hand, to check whether effects on variability and stability could be due to changes in movement amplitude and velocity. In case of significant MANOVA effects, to assess which characteristics determined the multivariate effects, univariate ANOVA using the same factors were performed.

## Results

[Dataset] Data available at Open Science Framework: Dataset Lumbar Movement Patterns repetitive seated reaching [30].

### Participants

With respect to age, sex, height, weight, and pain-related cognitions, participants with and without LBP were statistically comparable (Table 1) [9]. On the testing day the LBP-group experienced mild to moderate pain levels (NRS 2.4 (±2.1)) and moderate disability (ODI: 15.7 (±12.7)) [9]. Participants with LBP had a mean SBST score of 1.6 (±1.5) [9]. EBS values measured before the threat-condition and the reference trial were comparable between groups; The median of EBS$_{threat}$ in the back-healthy group was 2 (IQR 2.00–4.00) and in the LBP-group 3 (IQR 2.00–4.25) (U = 363, p = 0.19). The median of EBS$_{reference}$ in the back-healthy group was 2 (IQR 1.00–3.00) and 2 in the LBP-group (IQR 1.75–4.00) (U = 376, p = 0.26) [9]. The Wilcoxon signed-rank test indicated that EBS$_{threat}$ was significantly higher than EBS$_{reference}$ (Mean difference = 0.85, Z = -3.3, p<0.001). The EBS-scores between the two conditions were moderately correlated (r$_{spearman}$ = 0.598, p≤0.001). For further analyses participants were classified to high or low EBS-group, based on the dichotomized EBS-scores assessed prior to the first trial, because in the LBP-group, these scores were most strongly correlated with the lumbar movement patterns, with r$_{spearman}$ ranging from 0.36 (p = 0.05) to 0.54 (p≤0.00).

### Lumbar movement patterns

MeanSD, CyclSD, Mean_Amplitude and Velocity, were skewed and therefore log transformed. A main effect of threat was found on lumbar movement pattern characteristics, without interactions of group or EBS (Wilk's Λ = 0.632, F(5,52) = 6.068, p≤0.000; Table 2). Specifically, MeanSD$_{flexion-extention}$, CyclSD and LDE were higher in the threat-condition than in the reference-condition (Table 2 and Fig 3). Partial eta squared values for all three variables indicated large effects (Table 2). MeanSD$_{lateral-bending}$ and MeanSD$_{axial-rotation}$ were also higher in the threat condition, but not significantly (Table 2 and Fig 3).

The MANOVA also revealed an interaction effect of group and EBS on lumbar movement pattern characteristics (Wilk's Λ = 0.773, F(5,52) = 3.062, p≤0.017) (Table 2). This interaction was significant for MeanSD$_{axial-rotation}$ (p = 0.017), MeanSD$_{lateral-bending}$ (p = 0.003) and CyclSD (p = 0.039) (Table 2). Inspection of the data (Fig 4) showed that for spatial variability, this interaction mainly reflected high variability in the LBP participants with high EBS.

Postural threat also affected Mean_Amplitude$_{\_flexion-extension}$ and Velocity, without interactions with group or EBS (Wilk's Λ = 0.746, F(4,53) = 4.521, p≤0.003; Table 2). Particularly, Mean_Amplitude$_{\_flexion-extension}$ was higher (p = 0.006) and Velocity was lower (p = 0.004) in

**Table 2. Results of univariate ANOVAs with threat, group and high/low EBS$_{trial\_1}$ as independent and MeanSD, CyclSD and LDE (analysis 1), amplitude and velocity (analysis 2) as dependent variables.**

| Source | Dependent variable | | | | | F ratio | p value | Partial Eta Squared |
|---|---|---|---|---|---|---|---|---|
| **Analysis 1** | | **Reference Median (IQR)** | | **Postural threat Median (IQR)** | | | | |
| *Within-subject Threat* | MeanSD$_{flex-ext.}$ (degr.) | 1.66 (1.5–2.0) | | 2.03 (1.5–2.5) | | **19.565** | **0.000**\* | **0.259** |
| | MeanSD$_{axial-rot.}$(degr.) | 0.86 (0.7–1.0) | | 0.89 (0.7–1.0) | | 2.152 | 0.148 | 0.037 |
| | MeanSD$_{lat-bend.}$ (degr.) | 0.87 (0.7–1.0) | | 0.90 (0.7–1.0) | | 0.509 | 0.479 | 0.009 |
| | CyclSD (s) | 0.10 (0.8–0.1) | | 0.11 (0.1–0.1) | | **9.439** | **0.003**\* | **0.144** |
| | | *Mean (SD)* | | *Mean (SD)* | | | | |
| | LDE | 3.86 (0.2) | | 3.96 (0.3) | | **9.055** | **0.004**\* | **0.139** |
| | | **Back-Healthy & low EBS Median (IQR)**\*\* | **Back-Healthy & high EBS Median (IQR)**\*\* | **LBP& low EBS Median (IQR)**\*\* | **LBP& high EBS Median (IQR)**\*\* | | | |
| *Between-* | MeanSD$_{flex-ext.}$ (degr.) | 1.7 (1.5–2.2) | 1.9 (1.7–2.1) | 1.7 (1.5–1.9) | 1.9 (1.5–3.1) | 0.463 | 0.499 | 0.008 |
| *subjects* | MeanSD$_{axial-rot.}$ (degr.) | 0.9 (0.8–1.1) | 0.8 (0.7–1.0) | 0.7 (0.6–0.9) | 0.9 (0.8–1.1) | 6.022 | **0.017**\* | **0.097** |
| *Group*\* | MeanSD$_{lat-bend.}$ (degr.) | 1.0 (0.7–1.1) | 0.8 (0.7–1.0) | 0.8 (0.7–1.0) | 1.0 (0.9–1.4) | 9.745 | **0.003**\* | **0.148** |
| *EBS* | CyclSD (s) | 0.1 (0.1–0.1) | 0.1 (0.1–0.1) | 0.1 (0.1–0.2) | 0.1 (0.1–0.1) | 4.475 | **0.039**\* | **0.074** |
| | | *Mean (SD)* | *Mean (SD)* | *Mean (SD)* | *Mean (SD)* | | | |
| | LDE | 3.9 (0.2) | 4.0 (0.3) | 3.9 (0.3) | 3.9 (0.3) | 0.607 | 0.439 | 0.011 |
| **Analysis 2** | | **Reference Median (IQR)** | | **Postural threat Median (IQR)** | | | | |
| *Within-* | Mean_Ampl.$_{flex-ext.}$ (degr.) | 20.4 (13.1–26.4) | | 21.5 (17.1–27.0) | | **8.074** | **0.006**\* | **0.126** |
| *subject* | Mean_Ampl.$_{axial-rot.}$(degr.) | 8.2 (5.9–12.1) | | 8.1 (6.2–12.6) | | 1.561 | 0.217 | 0.027 |
| *Threat* | Mean_Ampl.$_{lat-bend.}$(degr.) | 7.9 (5.2–11.3) | | 8.4 (5.9–11.6) | | 1.315 | 0.256 | 0.023 |
| | Velocity (repetition/s) | 0.38 (0.33–0.42) | | 0.37 (0.33–0.38) | | **9.272** | **0.004**\* | **0.142** |
| *Between-* | Group | | | | | 1.356 | 0.262 | 0.093 |
| *subjects* | EBS | | | | | 2.363 | 0.065 | 0.151 |
| | Group\*EBS | | | | | 2.276 | 0.073 | 0.147 |

\*p = ≤0.05

\*\* Mean values of the Reference- and Threat-condition

the threat-condition than in the reference-condition. Partial eta squared values for Mean_Amplitude$_{flexion-extension}$ and Velocity indicated a medium and large effect respectively (Table 2).

## Discussion

The aim of the present study was to assess the effect of perceived postural threat on lumbar movement patterns. The results were in line with our hypothesis; postural threat increased variability and decreased stability of lumbar movement patterns. This effect was independent of the presence of LBP or level of EBS. Additionally, participants with LBP and high EBS, moved with higher spatial variability, regardless of threat. So, the interaction effect of EBS and LBP on

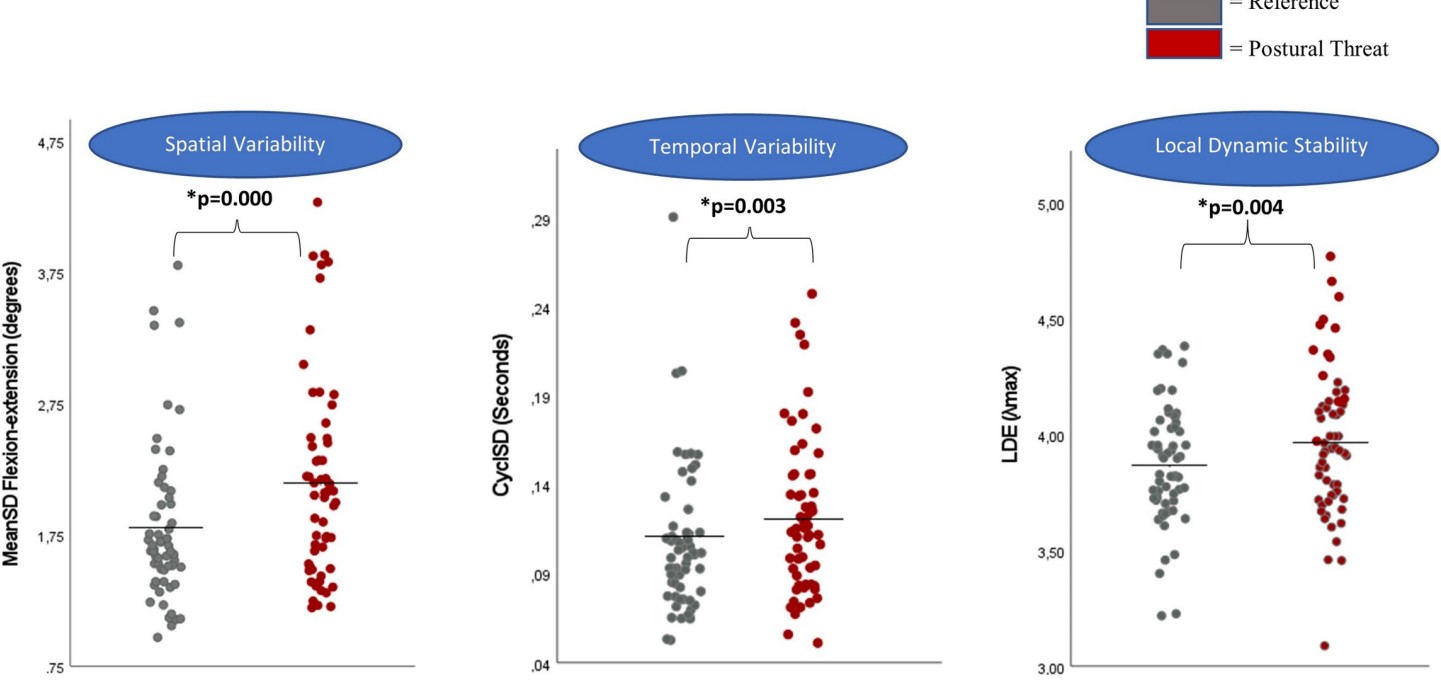

**Fig 3. Within-subject effect: Lumbar movement characteristics during repetitive seated reaching (40x) in the reference-trial and in a context of postural threat.** Dots represent individual participants, horizontal lines represent the group mean.

lumbar movement pattern characteristics we found in our previous study [9], was confirmed in the present study including both conditions: postural threat and the reference trial. These results are in line with the results of Ross et al. [17] who found more negative pain-related cognitions to be correlated with lower stability of lumbar movements in a situation with experimentally induced pain and with the neuromotor noise theory proposed by Van Galen et al. [31] which, more generally, states that arousal causes increased variability in motor outputs. A core tenet of this theory of Van Galen et al. [31], is that stochastic processes are present in the process of developing muscle force (neuromotor noise). Consequently, it is impossible to move without spatial variability (kinematic noise). This neuromotor noise is increased by stress (e.g., time pressure or a mental load) and co-contraction is a strategy to compensate for this increased neuromotor noise. Above a certain stress level, this neuromotor noise cannot be compensated anymore and the motor precision and the performance of the movement decrease. In terms of this neuromotor noise theory, during the threat condition, the signal-to-noise ratio exceeded the ability of the system to use stiffness (co-contraction) to compensate for the increased noise, reflected in an increased spatial variability [31].

A possible mechanism underlying increased variability due to arousal has been described by Ribot et al. [32] who found that mental computation and fist clenching caused increased variability of muscle spindle afference. Moreover, they showed that, in addition to the experimental task, individual characteristics influence the level of arousal or its effects. In some persons, the firing frequency of the muscle spindle afferents increased already while the instruction was given. Also, the duration of the increased sensitivity of the muscle spindles showed interindividual differences, varying from a decrease in sensitivity immediately after ending the experimental task to a continuing of this increased sensitivity for several minutes [32]. Muscle spindles contribute to proprioception, and impairments in the lumbar proprioception in people with LBP have been demonstrated [33]). Possibly, arousal, in our study

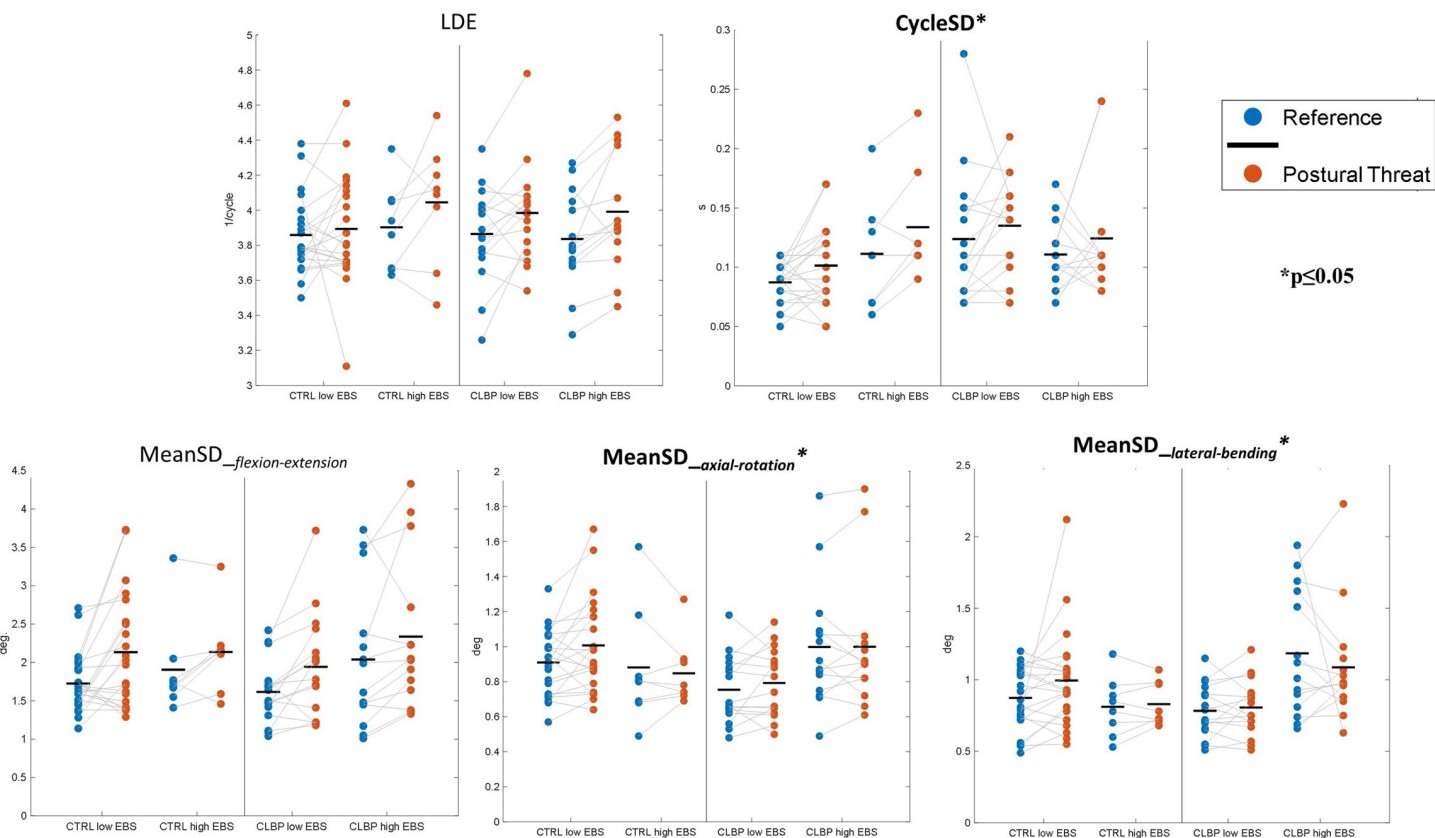

**Fig 4. Between-subject effect: Lumbar movement characteristics during repetitive seated reaching (40x) in the reference- and threat-conditions.** Dots represent individual participants grouped based on CLBP and EBS, horizontal lines represent the group means.

caused by pain-intensity measured at the day of testing, and perceived threat, increases variability of muscle spindles afference, which could then account for the effects on lumbar movement patterns observed here. In our results the influence of nociception or pain cannot be excluded through an increasing effect on variability and/or the EBS-value. Despite the absence of correlations between lumbar movement patterns and patient-reported pain-intensity at the day of testing in the LBP-group, the pain-intensity was higher in the subgroup with high EBS (median 3 (IQR 1–4.5) compared to low EBS (median: 1 (IQR 0–2.72), U = 61.5, p = 0.03). Ross et al. demonstrated that nociception decreased local dynamic stability of trunk movement [17].

We speculate that higher muscle spindle reflex gains, observed in patients with LBP and especially in those with negative pain-related cognitions are effective in dealing with external perturbations [34]. At the same time higher muscle spindle reflex gains, may amplify noise in spindle inputs (local muscle strain and strain rate) and as a consequence increase movement variability and decrease local dynamic stability.

The effect of perceived postural threat was mainly seen in movements around the primary movement axis (MeanSD$_{flexion-extension}$). Fig 4 shows that the back-healthy participants with low EBS showed increased MeanSD$_{lateral-bending}$ and MeanSD$_{axial-rotation}$ during the postural threat condition, but the LBP participants with high EBS did not. The values of the LBP participants with high EBS were fairly high compared to the other subgroups in both conditions. The lack of change in variability of movements around the secondary axes with postural threat in this group could thus by caused by a ceiling effect and seems to have damped the overall

effect of postural threat on these outcomes. We are not sure why these between-subject effects are observed for movement around the secondary axes. An increase in variability, only in the non-primary axes, with low back pain has also been observed in other studies [17, 35]. One might suggest that task constraints limit the scope to vary movements around the primary axis. An alternative explanation could be that an increased feedback gain (in the LBP group with high EBS) increases movement variability around the secondary axes more, because intrinsic stiffness is lower around these axes. However, the effect of threat was observed for movement around the primary axis, which would contradict both of these explanations.

In our study the threat-condition resulted in slower movements, lower local dynamic stability and more spatial variability. Asgari et al. [36] found that imposed slower flexion-extension trunk movements resulted in more variability but higher local dynamic stability. This difference in responses suggest that effects on our study were not mediated solely by slowing of movement.

As in our previous study, we found an interaction effect of group and EBS on lumbar movement patters. This interaction may reflect different values attributed to EBS, although we must be cautious in interpreting high and low EBS as a meaningful difference in perceived threat, given the lack of published measurement properties of the EBS-tool. In the subgroup LBP and high EBS, the experimental task may have activated pain-related cognitions, due to the increased risk for pain and injury, reflected in a higher EBS value, whereas an expectation of high strain may be more neutral to participants without LBP and the participants with LBP and low EBS. This indicates that, to assess task-specific perceived threat, additional questions may be needed concerning the value (threat, anxiety) attributed to the experimental task.

We focused in this study on the task-specific EBS instead of the general measures for pain-related cognitions, based on the lack of correlations between the lumbar movement patterns and general measures of pain-related cognitions in our previous study [9]. To investigate whether this was also the case in this study, we checked for correlations between the lumbar movement patterns characteristics in the threat condition on one hand and the Pain Catastrophizing Scale (PCS) the Pain Anxiety Symptoms Scale (PASS) on the other hand. In line with our previous study, no significant correlations were found.

We found an effect of threat on MeanSD and CycleSD, linear measures of the magnitude of the variance in the lumbar movement patterns and also on the LDE, a non-linear measure of the structure of the variance in movement patters. However, between-subject effects (the interaction of group and EBS) were found for linear measures only. This is in line with the conclusion of Saito et al. that linear measures reflecting the magnitude of the variance are more sensitive to pain-related cognitions than non-linear measures reflecting the structure of the variance [37]. A possible explanation could be that LDE is a less sensitive measure than MeanSD, despite being more reliable, as demonstrated in our previous study [13]. LDE had a smaller coefficient of variation (CV) than MeanSD. At the same time, the within-session intra-class correlation coefficient (ICC) of LDE was lower than that of MeanSD$_{flexion-extension}$. This implies that the variance of the LDE between participants was lower compared to that of MeanSD$_{flexion-extension}$. This may be reflected in a lower responsiveness to threat, which caused an increase of 3% in LDE and 22% in MeanSD$_{flexion-extension}$. A strength of this study is the experimental manipulation of postural threat, which provided the opportunity to establish causality between perceived threat and changes in lumbar movement patterns. However, the clinical importance remains to be elucidated. Whether the increased variability in the participants with LBP and high EBS persists in the longer term and in another context is unknown. Additionally, the threat used was of an artificial nature, which most likely does not resemble daily-life situations that are perceived as threatening. Effect sizes of such threats may well be quite different. Additionally, to further understand the underlying mechanism of increased

variability during perceived threat, it is recommended in future research, to study the response to mechanical perturbations, and possible associations with pain-related cognitions. This could verify the presence of decreased motor precision and increased spatial variability on one hand and enhanced recovery after perturbations on the other hand, as simultaneous effects of increased spindle reflex gains. Moreover, a prospective cohort study would be needed to assess the role of increased variability and decreased stability in the chronicity of LBP. It has not escaped our notice that the participants with LBP in this study had mild disability, while globally most disability is caused by 28 percent of the people with LBP, classified to the (most) severe categories [1]. Including patients from this severe category could further enhance the understanding of the association between pain-related cognitions and motor control, and is therefore recommended in future scientific research.

Postural threat resulted in increased variability and decreased stability of lumbar movement patterns, independent of the presence of low back pain (LBP) or level of 'Expected Back Strain' (EBS). Among participants with LBP, those with a higher level of task-specific EBS showed higher spatial variability of lumbar movement patterns, regardless of postural threat. These results suggest that perceived postural threat may underlie changes in motor behavior due to LBP.

## Supporting information

**S1 File. Description Expected Back Strain scale (EBS).**
(DOCX)

## Acknowledgments

The authors thankfully acknowledge the support provided in conducting the measurements by Julia Prent, Myrthe Konijnenburg, Sidney Jacobs, Florine Marinelli, Alie Weewer and Robyn Tesselaar.

## Author Contributions

**Conceptualization:** Meta H. Wildenbeest, Henri Kiers, Matthijs Tuijt, Jaap H. van Dieën.

**Data curation:** Meta H. Wildenbeest, Matthijs Tuijt.

**Formal analysis:** Meta H. Wildenbeest, Jaap H. van Dieën.

**Funding acquisition:** Jaap H. van Dieën.

**Investigation:** Meta H. Wildenbeest, Henri Kiers, Jaap H. van Dieën.

**Methodology:** Meta H. Wildenbeest, Henri Kiers, Matthijs Tuijt, Jaap H. van Dieën.

**Project administration:** Meta H. Wildenbeest.

**Resources:** Meta H. Wildenbeest, Henri Kiers, Jaap H. van Dieën.

**Software:** Meta H. Wildenbeest, Matthijs Tuijt, Jaap H. van Dieën.

**Supervision:** Henri Kiers, Matthijs Tuijt, Jaap H. van Dieën.

**Visualization:** Meta H. Wildenbeest.

**Writing – original draft:** Meta H. Wildenbeest.

**Writing – review & editing:** Henri Kiers, Matthijs Tuijt, Jaap H. van Dieën.

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
