## [Decision Letter · Decision Letter 0]

10 Oct 2022

PONE-D-22-25632Effect of postural threat on motor control in people with and without low back painPLOS ONE

Dear Dr. Meta Wildenbeest,

Thank you for submitting your manuscript to PLOS ONE. After careful consideration, we feel that it has merit but does not fully meet PLOS ONE’s publication criteria as it currently stands. Therefore, we invite you to submit a revised version of the manuscript that addresses the points raised during the review process.

We look forward to receiving your revised manuscript.

Kind regards,

Ravi Shankar Yerragonda Reddy, Ph.D

Academic Editor

PLOS ONE

- 'Associations of low-back pain and pain-related cognitions with lumbar movement patterns during repetitive seated reaching' (Gait & Posture, 2022, 91, 216-222; https://doi.org/10.1016/j.gaitpost.2021.10.032)

In your revision ensure you cite all your sources (including your own works), and quote or rephrase any duplicated text outside the methods section. Further consideration is dependent on these concerns being addressed.

5. Please expand the acronym “NWO” (as indicated in your financial disclosure) so that it states the name of your funders in full.

7. We note that you have stated that you will provide repository information for your data at acceptance. Should your manuscript be accepted for publication, we will hold it until you provide the relevant accession numbers or DOIs necessary to access your data. If you wish to make changes to your Data Availability statement, please describe these changes in your cover letter and we will update your Data Availability statement to reflect the information you provide.

Reviewers' comments:

Reviewer's Responses to Questions

**Comments to the Author**

1. Is the manuscript technically sound, and do the data support the conclusions?

Reviewer #1: Partly

Reviewer #2: Yes

2. Has the statistical analysis been performed appropriately and rigorously? 

Reviewer #1: Yes

Reviewer #2: Yes

3. Have the authors made all data underlying the findings in their manuscript fully available?

Reviewer #1: No

Reviewer #2: Yes

4. Is the manuscript presented in an intelligible fashion and written in standard English?

Reviewer #1: Yes

Reviewer #2: Yes

5. Review Comments to the Author

Reviewer #1: The manuscript “Effect of postural threat on motor control in people with and without low back pain” examines the effect of a perceived postural threat on lumbar spine kinematics in participants with and without low back pain. The manuscript is very well written and simple to follow. The data presented are valuable and the experiment itself is novel and appears to be a valid approach for inducing a perceived postural threat without requiring an actual perturbation. I would suggest, however, that the manuscript needs to be rewritten with a more conservative interpretation of the interaction effects that were observed between LBP status and the dichotomized high/low split for expected back strain (EBS).

I believe that the authors have interpretated of the main effect of ‘threat’ correctly. The data clearly demonstrate that the experimentally induced perception of a postural threat increases the temporal and spatial variability, and decreases the dynamic stability, of the lumbar movement pattern during the seated reaching task, independent of the experimental group (LBP vs. control) and the strain on the back that the participants expected from the task (EBS score).

The data also show interaction effects between group and EBS for some kinematic variables assessing movement variability. For these data, however, I do not agree fully with the authors’ interpretation that this is evidence of greater movement variability in participants with both LBP and high EBS than in the other three subgroups, regardless of the threat.

For the MeanSD variables in axial-rotation and lateral-bending, the authors’ interpretation has some merit. However, low EBS also appears to be associated with more variability in the control group. Because of this, as well as the limited amount of movement about these axes and the absence of any significant interaction effects for the primary axis of motion, I would be very hesitant to interpret these interactions without a model that accounts for the opposite effects of EBS on the control and LBP groups. Some explanation would also be warranted for why these effects did not suffer from the same “ceiling effect” that is alluded to as an explanation for why no main effect of threat was found in these off-axis movement data (bottom of page 14). I would expect any centrally-mediated effect be most evident in the primary axis movement (unless a less flexible motor control system is not capable exerting the same level of control in all movement axes).

For CycleSD, the interaction seems to be driven by the lower variability in the control subgroup with low EBS, while the high EBS control group has scores that are comparable to the LBP group, regardless of EBS. This does not seem to fit with the general statement that “… this interaction mainly reflected high variability in the LBP participants with high EBS.” (pg. 13) or that “… participants with LBP and high EBS, moved with higher variability, regardless of threat.” (pg. 14).

Other minor comments:

Abstract, last sentence:

“These results suggest that perceived postural threat may underlie changes in motor behavior in patients with LBP”. This conclusion does not seem appropriate, given the lack of between-groups differences in the data.

Introduction, second paragraph:

“Established risk factors for the development of chronic LBP (CLBP) are psychological factors …” could be interpreted as these being the only risk factors. I would suggest: “Established risk factors for the development of chronic LBP (CLBP) include psychological factors …”

Methods:

Joint Kinematics – pg 8 - Were the marker data filtered after resampling the data to 100Hz. If not, why not? High frequency noise in these data can be amplified during the rigid body and Eular angle calculations.

Variability - pg 8 - Was a cubic spline also used to normalize the angle data to 101 samples for analysis of spatial variability (as for the other two data interpolations mentioned in the methods)?

Velocity variable - using s/repetition as a measure of velocity seems backwards to me. I would expect to see velocity as repetitions/s so that a lower velocity is reflected by a lower value.

Results:

Participants – first sentence “with respect to age, sex, length, …” should be “with respect to age, sex, height, …”

Lumbar Movement Patterns (pg. 13) – for consistency “The MANOVA also revealed an interaction effect of LBP and EBS …” should be “The MANOVA also revealed an interaction effect of group and EBS …”

Discussion:

Pg 14, second paragraph: “Possibly, arousal, in our study caused by pain and perceived threat, increases variability of muscle spindles afference …”. Given that the current study only manipulated threat as a variable, it would be better not to speculate on the role of pain in your findings. Alternatively, the fact that pain intensity was higher in the LBP/high EBS group should be referred to here (it is addressed on pg. 15).

Reviewer #2: Thank you for the opportunity to review this manuscript. The topic of negative pain-related cognitions as a factor driving altered movement patterns in individuals with persistent back pain is an important area of research. The authors clearly understand the difficulties with using clinical pain-related cognition questionnaires of trait characteristics to test the relationship between pain-related cognition and trunk movement. Further, they developed a novel protocol that presents a task-specific trunk postural threat and employed a task-specific measure attempting to capture the associated pain-related cognition. The following comments and suggestions are made with an eye toward strengthening the manuscript and the reader's ability to interpret their findings.

Intro:

In paragraph 3 of the introduction, you introduce “local dynamic stability”. However, it may not be clear to all readers clear what you are referring to. It appears this is a measure of motor control, but what is getting measured with this approach? Since you then go on to use it as a measure of the kinematics, it seems it deserves a little more in the introduction as to the specific construct of motor control you are measuring.

“A contributing factor to these conflicting results could be that pain-related cognitions were measured with questionnaires addressing pain-related cognitions as a trait” – It would be helpful to the reader if you provided examples of which questionnaires you are referring to.

In paragraph 3 you introduce the need to use a task-specific measurement tool be used to measure pain-related cognitions. You then go on to describe your prior work where task demand “imposed a high back strain”. It may be best to use the term “imposed threat of pain or reinjury, until you get to the methods sections where you introduce the “Expected Back Stain” tool.

“We chose repeated seated reaching because this movement is a more demanding task for

the back than walking, which was the experimental task in most previous studies.” This statement needs a reference.

Methods:

Subjects: Were all the LBP subjects experiencing pain at the time of testing? If not, how might this have impacted the spread of data and your interpretation of findings?

Did any of the back-healthy participants have a history of low back pain episodes?

Kinematics- based on the location of the reflective markers, is the motion you derived just lumbar spine motion?

EBS tool- Is this a measurement tool you developed and are there measurement properties established/published for the tool? The reference you use here does not support the measurement properties of this tool. What were the anchors for the scale or the descriptors used for the 0-10 levels? Given that you use this measure to represent pain-related cognition or task threat level and create groups based on participants' scores on the scale, its measurement properties and validated construct are important information for the reader. Does the cut score used, represent a real difference in perceived threat?

Motor control is quantified by variability in cycle duration, motion variability, and slope of the divergence curves. Given that all of your readers may not be biomechanists or engineers, can you add a brief explanation of what aspects of motor control these variables represent?

Statistical analyses:

Please describe the statistical tests that were used to determine a lack of differences between the groups based on demographics.

The second sentence in the statistical analyses paragraph is difficult to follow.

It appears you set the p values at .05 for each analysis. Given that the dependent variables appear associated, can you address any concerns for type 1 errors within the discussion section?

Results:

“Participants with and without LBP were comparable..” do you mean statistically not different?

The EBS dichotomized the groups, is the mean difference meaningful based on the descriptors you used for the scale. This should be addressed in the discussion.

Your figures 3 and 4 are well done. Thank you for sharing the full data with the reader.

Discussion:

The differences found between the reference and threat conditions do not numerically appear large, particularly when reviewing the figures. You have a previously published reliability stability on these variables which demonstrates moderate within-session reliability. This dataset should allow you to calculate minimal detectable differences which may assist in the further interpretation of the statistical differences you are reporting.

Please expand upon the “neuromotor noise theory” and further relate your findings to the theory.

In the second paragraph:

Does arousal equate to your EBS score?

You mention that you believe arousal creates a change in variability mediated by muscle spindle afference and provided the reference that a previous study demonstrated “mental computation and fist clenching” increased variability of muscle spindle afference. Can your further explain or clarify how “mental computation and fist clenching” and dependence up internal attitude found in that study are similar to threat-related arousal?

“Possibly, arousal, in our study caused by pain and perceived threat, increases variability of muscle spindles afference, which could then account for the effects on lumbar movement patterns observed here” I may have missed it, was a change in pain measured prior to the reference trial?

The sentence “We speculate that higher… local dynamic stability” is long and difficult to follow. It may help to break it apart.

“This is in line with the conclusion of Saito et al. that linear measures reflecting the magnitude of the variance are more sensitive to pain-related cognitions than non-linear measures reflecting the structure of the variance (31).” Could you expand more as to why this might be the case?

6. PLOS authors have the option to publish the peer review history of their article (what does this mean?). If published, this will include your full peer review and any attached files.

Reviewer #1: No

Reviewer #2: No

---

## [Author Response · Author response to Decision Letter 0]

21 Nov 2022

Plos One

Emily Chenette

Public library of science

Cambridge

Great Britain

Date: November 21, 2022

Subject: submission revision

Dear prof. Chenette,

We would like to submit the revision of our manuscript entitled: ‘Effect of postural threat on motor control in people with and without low back pain’.

We have made the following adjustments;

• We have added a ‘Response to Reviewers’ in which we have responded to each point raised by the reviewers. Additionally, we have added a ‘Revised manuscript with track changes’ and a unmarked version, labeled as ‘Manuscript’. 

• We have addressed minor occurrence of overlapping text with the following previous publication(s),: 'Associations of low-back pain and pain-related cognitions with lumbar movement patterns during repetitive seated reaching' (Gait & Posture, 2022, 91, 216-222; https://doi.org/10.1016/j.gaitpost.2021.10.032). 

• We have expanded the acronym “NWO” in the financial disclosure so that it states the name of your funders in full.

• We have made our data available at Open Science Framework: Dataset Lumbar Movement Patterns repetitive seated reaching. https://doi.org/10.17605/OSF.IO/9XSNP

• We have added in the manuscript that the participants provided their written consent. 

• We have matched the grant information provided in the ‘Funding information’ and ‘Financial Disclosure’.

• We have added the heading ‘Acknowledgments’ in the manuscript.

• We have added supplementary material (suppl. Material_1)

Yours sincerely,

Meta (M.H.) Wildenbeest

---

## [Decision Letter · Decision Letter 1]

2 Dec 2022

PONE-D-22-25632R1Effect of postural threat on motor control in people with and without low back painPLOS ONE

Dear Dr. Meta Wildenbeest

Thank you for submitting your manuscript to PLOS ONE. After careful consideration, we feel that it has merit but does not fully meet PLOS ONE’s publication criteria as it currently stands. Therefore, we invite you to submit a revised version of the manuscript that addresses the points raised during the review process.

We look forward to receiving your revised manuscript.

Kind regards,

Ravi Shankar Yerragonda Reddy, Ph.D

Academic Editor

PLOS ONE

Journal Requirements:

Reviewers' comments:

Reviewer's Responses to Questions

**Comments to the Author**

1. If the authors have adequately addressed your comments raised in a previous round of review and you feel that this manuscript is now acceptable for publication, you may indicate that here to bypass the “Comments to the Author” section, enter your conflict of interest statement in the “Confidential to Editor” section, and submit your "Accept" recommendation.

Reviewer #1: (No Response)

Reviewer #2: (No Response)

2. Is the manuscript technically sound, and do the data support the conclusions?

Reviewer #1: Yes

Reviewer #2: Yes

3. Has the statistical analysis been performed appropriately and rigorously? 

Reviewer #1: Yes

Reviewer #2: Yes

4. Have the authors made all data underlying the findings in their manuscript fully available?

Reviewer #1: Yes

Reviewer #2: Yes

5. Is the manuscript presented in an intelligible fashion and written in standard English?

Reviewer #1: Yes

Reviewer #2: Yes

6. Review Comments to the Author

Reviewer #1: The manuscript “Effect of postural threat on motor control in people with and without low back pain” presents valuable data using a novel experimental model. I am satisfied with the changes that the authors have made based on my comments (and those of the other reviewer), as well as with the arguments provided where changes were not made.

I have only a few minor comments but will leave it to the discretion of the authors and editor how (or whether) to address these.

All page and line numbers refer to the newly-edited manuscript version without tracked changes.

1) Page 3, line 72 – I would not abbreviate ‘resp.’. I assume that this should read: “… variability and local dynamic stability (LDS) as, respectively, linear and non-linear outcome measures …”

2) Page 3, line 74 – ‘Temperal variability’ should read ‘Temporal variability’

3) Page 4, line 77 - ‘… after infinitesimally perturbations …’ I assume should be ‘… after infinitesimally small perturbations …’. I would, however, suggest that ‘infinitesimally’ may be an exaggeration and that ‘… after very small perturbations …’ would be more appropriate.

4) Page 4, line 85 – I am not sure that pain-related cognitions can be considered to be ‘character traits’. I would suggest writing simply “… pain-related cognitions are dynamic and mutable, influenced by the task at hand and its current context.”

5) Page 6, line 139 (and elsewhere throughout the manuscript) – I believe that both ‘T’s and the ‘B’ should be capitalized when referring to the ‘STarT Back’ screening tool. The acronym used on the same line (SBST) also appears as (STBST) elsewhere in the manuscript.

6) Page 6, line 142 – both uses of ‘indicated’ do not appear to fit the context of the sentence. I would suggest either “… with a minimum score indicating …” or “… for which a minimum score indicated …”).

7) Page 15, line 47 – at the end of the line ‘… varying form a …’ should read ‘… varying from a …’

8) Page 17, lines 101-103 – I would suggest that the more relevant measurement property be emphasized. For example, "A possible explanation could be that LDE is a less sensitive measure than MeanSD, despite being more reliable, as demonstrated in our previous study."

9) Page 17, lines 103 – the acronym for ‘coefficient of variation’ is written in lowercase (cv) while all other acronyms are in uppercase. I would suggest using (CV) throughout.

10) Throughout the manuscript, the use of commas does not seem to follow any convention that I am familiar with. Given that more than one convention exists, however, I would suggest that the editor (or copy editor) suggest any changes that might be required based on the guidelines of the journal.

Reviewer #2: Thank you for addressing our prior comments.

We would like to make an additional suggestion. Since you acknowledge the limitations of the EBS instrument, "The cut score used is based on the median and is an arbitrary choice. We do not suggest that this reflects a hard difference between perceiving much or little threat." It would make it more transparent to the reader that you are not suggesting the high vs. low EBS reflects a meaningful difference in perceived threat if you acknowledge this as a point to be considered when interpreting the significant interaction reported for LBP and High EBS related to spatial variability.

7. PLOS authors have the option to publish the peer review history of their article (what does this mean?). If published, this will include your full peer review and any attached files.

Reviewer #1: **Yes: **Richard Preuss

Reviewer #2: No

---

## [Author Response · Author response to Decision Letter 1]

9 Dec 2022

Dear Reviewers, 

Thank you for your constructive comments. We have responded to each of your comments in the document 'Response to reviewers Revision2' 

Yours sincerely,

Meta (M.H.) Wildenbeest

---

## [Editor Report · Decision Letter 2]

4 Jan 2023

Effect of postural threat on motor control in people with and without low back pain

PONE-D-22-25632R2

Dear Dr. Meta Wildenbeest

We’re pleased to inform you that your manuscript has been judged scientifically suitable for publication and will be formally accepted for publication once it meets all outstanding technical requirements.

Kind regards,

Ravi Shankar Yerragonda Reddy, Ph.D

Academic Editor

PLOS ONE

<quillbot-extension-portal></quillbot-extension-portal>

---

## [Editor Report · Acceptance letter]

10 Jan 2023

PONE-D-22-25632R2 

Effect of postural threat on motor control in people with and without low back pain 

Dear Dr. Wildenbeest:

I'm pleased to inform you that your manuscript has been deemed suitable for publication in PLOS ONE. Congratulations! Your manuscript is now with our production department. 

Kind regards, 

on behalf of

Dr. Ravi Shankar Yerragonda Reddy 

Academic Editor

PLOS ONE